# Evaluation of Surface Crack Development and Soil Damage Based on UAV Images of Coal Mining Areas

**Fan Zhang** [1], **Zhenqi Hu** [2,*], **Yusheng Liang** [1] **and Quanzhi Li** [1]

1   School of Geoscience and Surveying Engineering, China University of Mining and Technology, Beijing 100083, China
2   School of Environment Science and Spatial Informatics, China University of Mining and Technology, Xuzhou 221116, China
*   Correspondence: huzq@cumtb.edu.cn

**Abstract:** Coal mining is necessary for the development of society but at the same time causes ecological damage that must also be repaired based on science. In the arid and semi-arid regions of northwest China, surface cracks are one of the major geo-environmental problems caused by coal mining, and studies are urgently needed to determine how to effectively repair them in a scientific manner. The rapid development of unmanned aerial vehicle (UAV) remote sensing technology in recent years has resulted in a good source of data for acquiring feature information on surface cracks. Existing studies mainly focus on high-precision crack extraction methods, and there are few studies on the methods for evaluating cracks. However, clarifying the degree of cracks requiring repair and what repair measures are required through scientific and reasonable evaluation methods is necessary to formulate effective crack repair and land reclamation plans. Given these considerations, in this study, the degree of both crack development and soil damage were evaluated based on the crack extraction results of UAV images. Based on the results of indoor experiments and field measurements, the grading criteria for the degree of crack development and soil damage were constructed. Crack density was used as the evaluation index for the degree of crack development (slight: <0.4%, moderate: 0.4–2%, severe: >2%). The distance between soil and cracks was the basis of the evaluation index for the soil damage degree (severe damage area: <0.6 m; slight damage area: 0.6–1.2 m; no obvious damage area: >1.2 m). Through the results from evaluating the degree of both crack development and soil damage in the study area, it was found that the degree of crack development was mainly moderate and located in the northern crack zone of the study area, with the cracks and damaged soil showing a striped pattern in the east-west direction. Combining the evaluation results of crack development and soil damage, the ecological restoration model of "natural restoration + crack filling + water supplementing + vegetation planting" is proposed. We conclude that crack repair should be applied in areas where moderate and severe cracks have developed, whereas soil repair should target the soil within 1.2 m of the cracks in the above area. This study is the first attempt to construct and evaluate the classification criteria of crack development degree and soil damage degree from the perspective of cracks and soil, and the results are of guiding significance for land reclamation in mining areas.

**Keywords:** UAV images; surface crack; crack development degree; soil damage evaluation





## 1. Introduction

As an important primary energy source in China, coal plays an important role in the country's economic development and will continue to be mined as a major source of energy in the coming decades [1]. However, coal mining has also led to significant ecological and environmental issues [2]. It is located in the border region between the Maowusu Desert and the Loess Plateau in northern Shaanxi, in the arid and semi-arid region of northwest China. With its undulating landscape, crisscrossed loess gullies, and sandy

winds, it is a typical ecologically fragile location [3]. Large-scale coal seam development and use will unavoidably result in more severe issues with subsidence, water resources, and the environment than in the plains, as well as a number of geological environmental issues in mines [4]. Surface cracks are one of the most major geological environmental issues brought on by coal mining in this region, resulting in the degradation of arable land, increased soil erosion, destruction of vegetation, damage to subsurface pipelines, and deformation of buildings [5–7], which greatly hampers the management and repair of coal mining regions [8]. Therefore, a problem that needs to be studied urgently at present is how to quickly and easily acquire high-precision information of the quantitative features and distribution of surface cracks in coal mining subsidence areas, followed by the scientific and reasonable evaluation of surface cracks to provide data support and guarantee for land reclamation work [9].

Today, field surveys, radar detection methods [10,11], satellite remote sensing imaging [12], and UAV imaging [13] are the main methods used to gather information about surface cracks. Field studies, however, are expensive and ineffective [14], radar detection is frequently used to monitor landslides [15], and the low resolution of satellite remote sensing photos makes it difficult to detect minute fissures [16]. Due to their excellent resolution, efficiency, ease, and low cost, unmanned aerial vehicles (UAVs) have been widely exploited as an appropriate data source for secure information collecting [17]. Edge detection [18], threshold segmentation [19], object-oriented [20,21], and manual visual interpretation [14] are the primary techniques for extracting cracks from UAV images. However, surface vegetation and other feature types have a significant negative impact on techniques such as edge detection and threshold segmentation, which can result in a large number of inaccurate pixels and low crack extraction accuracy. The object-oriented methodology is as time-consuming as the manual visual interpretation method, since it involves numerous steps and uses spectral features to examine geometric features, linear features, fractal dimensions, etc. At the same time, some scholars extract cracks based on artificial intelligence and deep learning methods [4,9]. This method has significant advantages in the efficiency and results of crack extraction. However, due to the complex geological environment in the mining area and the dense vegetation on the surface, how to effectively avoid the interference of vegetation on the crack extraction results remains to be further studied. The appropriate depth learning algorithm will be able to obtain more accurate crack extraction results, but an inappropriate algorithm will not only increase the time cost but also be vulnerable to the interference of vegetation. In addition, the quantitative measurement and statistics of crack feature information have also been accomplished by certain researchers using image processing and pattern recognition techniques, and preliminary investigations have shown some promising results [22,23]. These studies, however, primarily concentrate on the fundamental study of how to extract information about cracks and their characteristics from photographs. Moreover, there is no established technique for assessing the damage caused by surface cracks in coal mining subsidence sites; therefore, crack repair and land reclamation plans continue to be insufficient. In the study of Yusheng Liang et al. on the evaluation of surface crack damage based on kernel density estimation, only the development density of the cracks was assessed without taking into account the damage to the soil caused by the fractures, which made it insufficient for the needs of land reclamation and treatment programs [24]. Therefore, there is an urgent need to propose a scientific and reasonable surface crack damage evaluation method as guidance for carrying out efficient and reasonable ecological restoration work.

The first two issues to be clarified in crack repair and land reclamation are whether crack development is severe and where the damaged soil in the coal mining subsidence area is located. Therefore, to solve the above problems, this study uses the crack extraction results of UAV images as a basis for evaluating the degree of both crack development degree and soil damage. On the one hand, evaluation of the crack development degree is used to answer whether the cracks are serious and need to be repaired. On the other hand, the evaluation of soil damage degree is used to answer where manual treatment

measures are required for damaged soil. It aims to provide scientific and reasonable data support and a theoretical basis for crack repair and land reclamation through the above evaluation results. In the evaluation, the grading criteria are constructed by combining the theoretical basis of crack self-healing [25] and the recommendations of crack management experts. Through indoor experiments and field measurements, we classify the degree of crack development into three levels of slight development, moderate development, and severe development, and classify the degree of soil damage in an area into severe damage, slight damage, and no obvious damage according to the effect of cracks on soil moisture. The specific research methods are described in detail in the following.

The rest of the article is organized as follows. Section 2 presents the materials and methods, Section 3 presents the results, Section 4 provides the discussion, and the conclusions are drawn in the final section.

## 2. Materials and Methods

### 2.1. Data Source and Research Methods

The study area is located in the coal mining surface subsidence area of Yulin City, northern Shaanxi Province, China. In this study, the surface cracks caused by coal mining are the study object. The parameter information of the UAV image data is shown in Table 1. Figure 1 shows the UAV image of the study area, which has a size of 28,000 × 20,000 pixels. The study area is an arid and semi-arid region, and the surface soil types are mainly loess and sandy.

**Table 1.** UAV image data parameter information.

| Parameters | Value |
| :---: | :---: |
| Data type | Visible light image |
| Flight date | 1 July 2021 |
| Flight height | 50 m |
| UAV model | M210RTK |
| Camera model | DJI M100_X3 |
| Focal length | 3.6 mm |
| Ground spatial resolution (GSD) | 1.3 cm |

### 2.2. Research Methods

In this study, the methods for evaluating the degree of both crack development and soil damage were studied and discussed, and the technical roadmap is shown in Figure 2. We divide the whole research process into three stages.

Stage one is data acquisition. Images are acquired through the layout of image control points and aerial photography of UAV. The scope of the study area is small, so we set up 9 ground control points in this area and use RTK to measure the set ground control points to get the actual coordinates of the control points. Pix4dMapper software is used to splice and preprocess the original image of UAV.

Stage two is crack extraction. The sub-image datasets is constructed and the best crack extraction method is selected. Then the crack extraction results are further processed, such as outlier removal, hole filling, etc. Detailed methods are described in Section 2.3.

Stage three is crack evaluation. The grading criteria were constructed and evaluated from two aspects: the crack development degree and the soil damage degree, respectively. Finally, a scientific and reasonable plan for crack repair and land reclamation is formulated based on the evaluation results. Detailed methods are described in Sections 2.4 and 2.5.

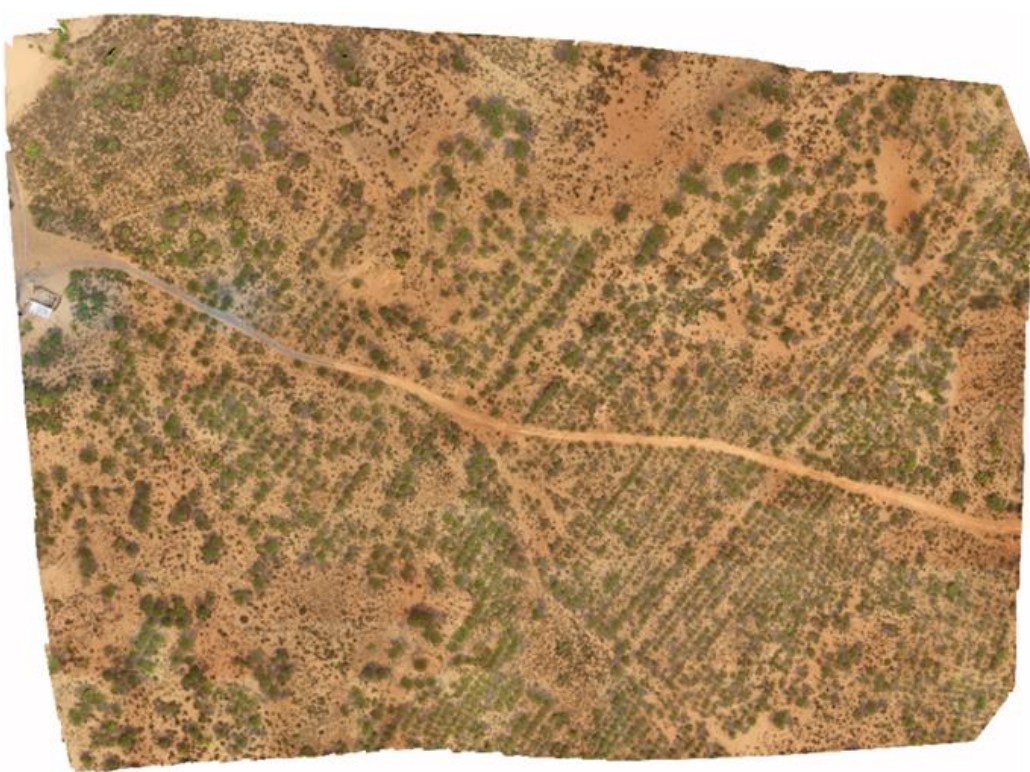

**Figure 1.** The UAV image synthesized through RGB bands.

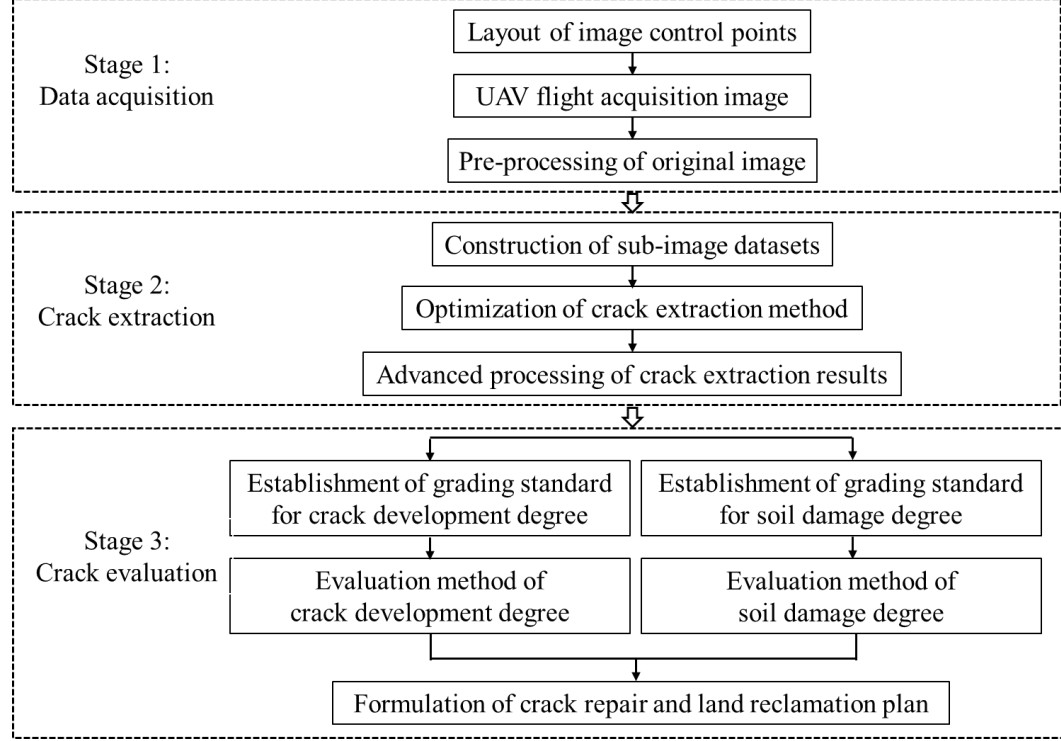

**Figure 2.** Technology roadmap for the research methodology.

*2.3. Crack Extraction Method Based on Machine Learning*

The object of this study is mainly surface cracks in coal mining subsidence areas, which represent a form of nonlinear mechanical hazard coupled with the movement

of overlying rock layers and deformation of topsoil layers in the mined area after coal mining [1]. High-precision and efficient crack extraction methods based on UAV images is currently a hot research topic for scholars. The geological environment of the mine area is complex, given the undulating surface, a scattered vegetation distribution, and especially the similar spectral color characteristics of withered vegetation and surface cracks, which causes trouble for crack extraction. How to effectively avoid interference from other feature types, such as vegetation, is a technical difficulty that needs to be urgently solved for high-precision crack extraction. In recent years, as the requirement for crack extraction accuracy increases, traditional methods such as edge detection and threshold segmentation are unable to meet the demand. More and more scholars are gradually applying artificial intelligence methods for image recognition and crack detection with good results. Hoang et al. found that the machine-learning approach of SVM is significantly better than RF and KNN in their study of road crack extraction [26]. Chen et al. obtained high-accuracy crack extraction results by combining local binary patterns (LBP) and support vector machine (SVM) [27]. Li Gang et al. used a deep learning approach to design an automatic bridge crack identification system [28]. Zhao et al. extracted the cracks by thermal infrared images and obtained the optimal image acquisition time [29]. In order to reduce the interference of vegetation on crack extraction, Gruszczynski et al. presented an approach to the problem of minimizing the impact of low vegetation on the accuracy of a UAV-derived DEM, based on the use of a deep neural network (DNN) [30]. Cwiakala et al. reduced the impact of vegetation through the selection of shooting time and image segmentation in the use of UAV photogrammetry to monitor surface deformation [31]. Puniach et al. propose a workflow for automatic determination of the field of horizontal displacements caused by underground mining with the use of ultra-high resolution methodologies, which can also effectively avoid the impact of vegetation [32]. Zhang et al. obtained better crack extraction accuracy by cutting the complete UAV image to obtain small sub-images for crack extraction, effectively avoiding vegetation from interfering with the crack extraction results, and obtained the best cell scale and extraction method through comparative analysis [9]. Therefore, based on the above research, this study proposes a surface crack extraction method based on machine learning, as shown in Figure 3. The steps are shown as follows:

Firstly, the UAV image of the study area is cut into $50 \times 50$ sub images using the image-cutting method. Secondly, two datasets (bare ground dataset and vegetation dataset) are constructed based on background information of the sub-images. The sub images of the bare ground dataset contain only bare ground in addition to cracks, and crack extraction is a binary classification. In addition to cracks, the sub images in the vegetation dataset include other features such as bare ground and vegetation, and crack extraction is a multi-classification process. Thirdly, image enhancement of the sub images is performed though Laplace filtering, dimensionality reduction is achieved through principal component analysis (PCA), and image recognition of the sub images is performed using the support vector machines (SVM) machine learning method, and the sub images of the two datasets are divided into two datasets of crack and no crack. Fourth, sub images with cracks for crack extraction and sub images with no cracks are backgrounded. Fifth, all the processed images are re-stitched to obtain the complete crack extraction image for the study area. Sixth, the stitched images were processed using morphological methods such as isolated point removal, hole filling, and crack bridging to obtain the finely processed crack extraction results of the UAV images. In particular, it should be noted that the sub image size of $50 \times 50$ pixels is an empirical value that has been proven to be a better unit scale [9]. In different study areas, it can also be cut into different sub images sizes, depending on the actual situation.

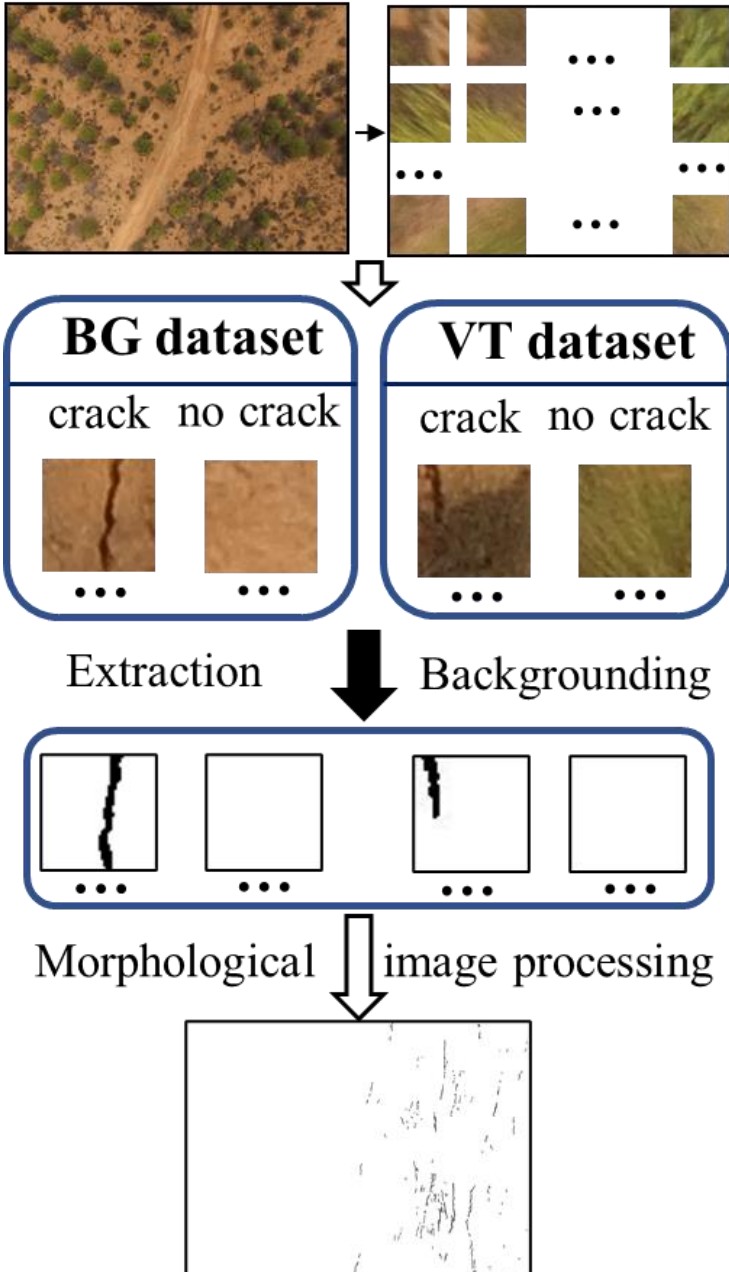

**Figure 3.** The steps of the crack extraction method based on machine learning.

*2.4. Grading Criteria Construction and Evaluation Method of Crack Development Degree*

The generation of surface cracks in coal mining subsidence areas is influenced by the location of underground coal mining workings, and presents special distribution characteristics at the surface. Therefore, if the cracks in the entire study area are studied as a whole, it will not be possible to obtain the distribution characteristics of the cracks. Therefore, to determine where the cracks are severely developed, it will not be possible to design a scientific and reasonable plan for crack repair and land reclamation. Given these considerations, in this study, grading criteria are constructed and an evaluation method for crack development degrees based on grids is proposed. Firstly, the whole study area is divided into small grids containing location and area attributes. Then, the degree of crack development in all grids is graded and evaluated to provide supporting data for subsequent crack repair and land reclamation. Considering the situation in the

construction conditions, the grid length was set to approximately 5 m on the advice of the crack management experts, and the grid size was 400 pixels in the UAV images.

Firstly, the grading criteria for the degree of crack development need to be constructed. Underground coal mining causes cracks in the soil of the surface subsidence area, which increases the evaporation area and infiltration channels for soil moisture and increases the evaporation intensity of soil moisture during non-rainfall periods, resulting in a large amount of rainwater infiltrating deep underground along the cracks during rainfall, which causes the surface moisture holding capacity to decrease [33]. Zhang Yanxu's study also showed that surface cracks caused a decrease in soil water content and that there was a significant negative correlation between crack density and soil water content, i.e., the higher the crack density, the lower the soil water content [3]. It is seen that the more severe the degree of crack development, the greater the effect of cracks on soil water content. Given these considerations, in this study, the grading criteria for the degree of crack development are constructed based on the results for the effect of different crack densities on the variation of soil water content. Since the study area is located in the arid and semi-arid region of western China, the soil types are mainly loess and sandy, so the experiments were conducted separately for loess and sandy soils. The experimental parameters are shown in Figure 4 and Table 2 below, where Figure 4a is sandy soil and Figure 4b is loess. The experimental procedure is as follows:

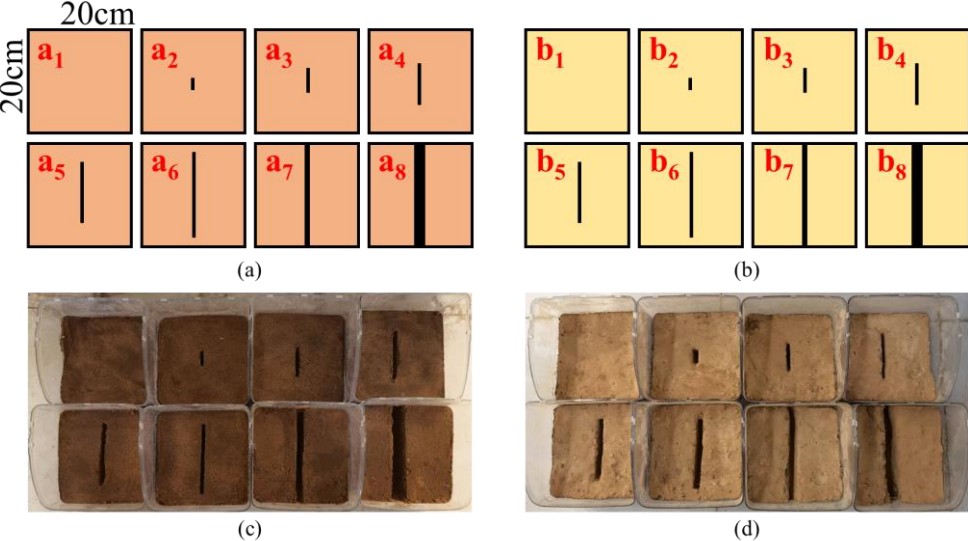

**Figure 4.** Soil water evaporation experiments for loess and sandy soil under different crack densities. (**a**) is a design diagram of different crack sizes on the sand soil, (**b**) is a design diagram of different crack sizes on the loess, $a_1$ and $b_1$ have no cracks, and $a_2$ to $a_8$ and $b_2$ to $b_8$ have different crack sizes.

**Table 2.** Design parameters for different crack densities.

| Number | a1, b1 | a2, b2 | a3, b3 | a4, b4 | a5, b5 | a6, b6 | a7, b7 | a8, b8 |
|---|---|---|---|---|---|---|---|---|
| Crack size/cm | $0 \times 0$ | $0.5 \times 3.2$ | $0.5 \times 6.4$ | $0.5 \times 9.6$ | $0.5 \times 12.8$ | $0.5 \times 16$ | $1.0 \times 20$ | $2.0 \times 20$ |
| Crack density | 0 | 0.40% | 0.80% | 1.20% | 1.60% | 2.00% | 5.00% | 10.00% |

(1) The loess and sandy soil were dried and placed in a square box of size 20 cm × 20 cm. Cracks were designed as shown in Table 2. Theno crack control groups a1 and b1 are shown in Figure 4a,b. (2) To each, 500 mL of distilled water is added, followed by weighing and recording for each group of experimental devices. They are then placed in the same environment, the moisture is allowed to naturally evaporate for 24 h, and they are then

reweighed to calculate the soil moisture evaporation in 24 h. (3) The 24-h soil moisture evaporation was used as a grading index to classify the crack development into three levels of slight development, moderate development, and severe development.

In particular, it should be noted that the crack density refers to the proportion of the crack area to the area of the region.

Based on the above criteria grading of crack development degree and the results of crack extraction from UAV images, the crack development degree in the study area is evaluated. Firstly, the crack extraction image is divided into 400 × 400 pixels (actually approximately 5.2 m × 5.2 m) small grid. Then the crack density in each grid is calculated, and all grids are classified into three levels slight development, moderate development, and severe development according to the grading criteria of the crack development degree. Finally, the results for evaluation of the crack development degree are analyzed and discussed.

### 2.5. Grading Criteria Construction and Method for Evaluation of Soil Damage Degree

The main environmental impact of the cracks in coal mining subsidence areas is reflected in the quality of the soil, so in addition to the evaluation of the crack development degree, it is also necessary to evaluate the soil damage degree. Surface cracks change the distribution of the soil water content in the surrounding area, threatening the growth of vegetation. Ma et al. found that the soil water content was lower closer to the crack [34]. Zhang et al. found that larger cracks (5–10 cm in width) and smaller cracks (1–3 cm in width) had a significant effect on soil water content in the range of 60 cm and 30 cm from the crack, respectively [35]. Xu et al. found that cracks caused by coal mining did not have a significant effect on soil water content when the distance exceeded 120 cm [36]. It can be seen that the degree of soil damage decreases as the distance between the soil and the crack increases. Given these considerations, in this study, grading criteria are constructed, and a method for the evaluation of soil damage degree based on the buffer is proposes. We constructed the soil damage grading criteria based on the distance between the cracks and the surrounding soil and divided the damaged soil in the study area into severe damage areas, slight damage area, and no obvious damage areas. The aim is to clarify the degree and scope of soil quality in the study area affected by cracks.

In this study, soil damage grading criteria were constructed based on the variation in soil water content at different distances from the cracks. Since the width range of surface cracks in this study area is basically in the range of 5–10 cm, cracks with a width of approximately 8 cm were selected for the study. We selected one crack in each of the loess and sandy soil areas for measurement. The measuring tool for soil moisture content is the TDR (time domain reflectometry) Soil Moisture Sensor. The measurement depth is 10 cm. Soil moisture content was measured at 15 cm intervals within 180 cm from the crack. The soil water content at each distance was measured five times and averaged. Figure 5 shows a schematic diagram of the location of the soil water content measurements at different distances from the cracks.

Based on the above criteria grading of soil damage degree and the results of crack extraction from UAV images, the soil damage degree in the study area is evaluated. First, the crack extraction image is imported into ArcGIS and the cracks are extracted as line files. Then, we use the "buffer" tool in ArcGIS to create buffers for severely damaged areas and slightly damaged areas based on the grading criteria of soil damage degree. Finally, the results for evaluation of soil damage degree are analyzed and discussed.

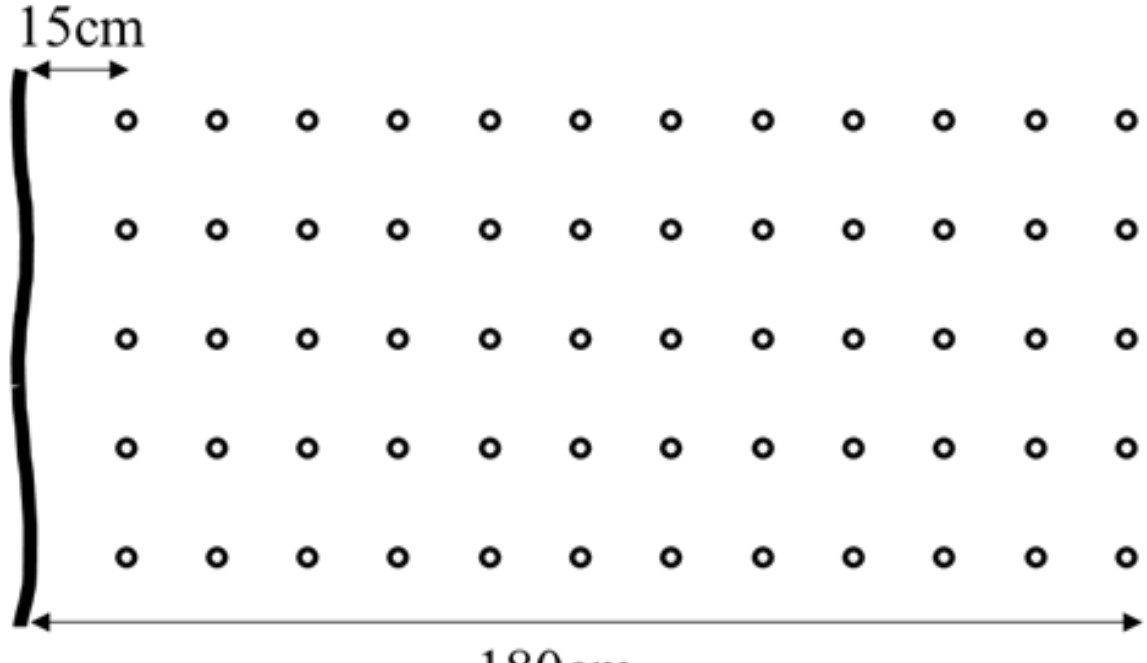

**Figure 5.** The location of the soil water content measurements at different distances from the cracks.

### 3. Results

#### 3.1. Surface Crack Extraction Results

Figure 6 shows the crack extraction results of the UAV image. It can be seen that the crack extraction results obtained by the method proposed in this article can effectively avoid the interference of other features such as vegetation and obtain more accurate crack extraction results. This can provide reliable data support for the subsequent quantitative calculation of crack features information, and, thus, guarantee the scientific and reliability of the crack evaluation.

#### 3.2. The Results of the Construction of Grading Criteria and Evaluation for the Crack Development Degree

Cracks accelerate the evaporation of soil moisture, and the more serious the development of cracks, the faster the evaporation of soil moisture. Therefore, in this study, the degree of crack development was inverted by the soil moisture evaporation in 24 h, and then the grading criteria were constructed. Tables 3 and 4 shows the experimental results for soil moisture evaporation in 24 h under different crack densities in loess and sandy soil, respectively. Figure 7 shows the soil moisture evaporation in 24 h from loess and sandy soils with increasing crack density. It can be seen that the magnitude of the soil moisture evaporation variation significantly increased at crack densities of 0.4% and 2.0% in 24 h, respectively. Therefore, the results for the construction of grading criteria for the degree of crack development in this study are shown in Table 5. A crack density of less than 0.4% is slight development. Crack density between 0.4% and 2% is moderate development. Crack density greater than 2% is severe development.

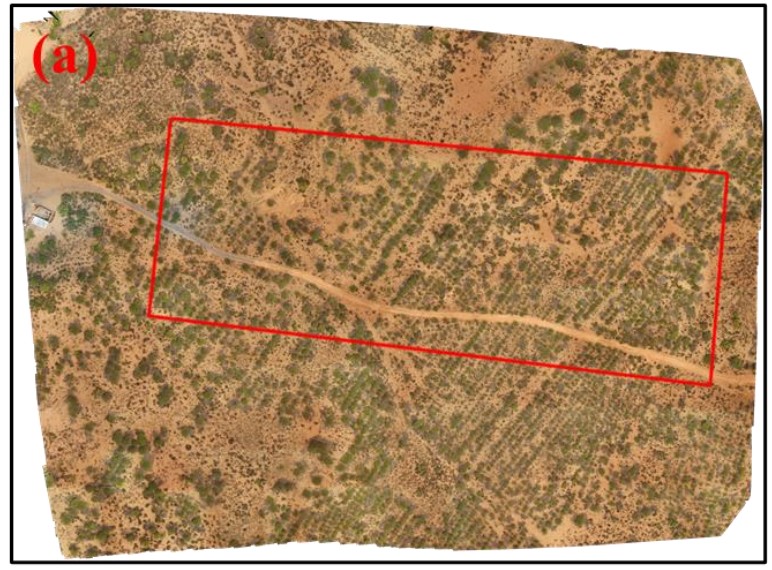

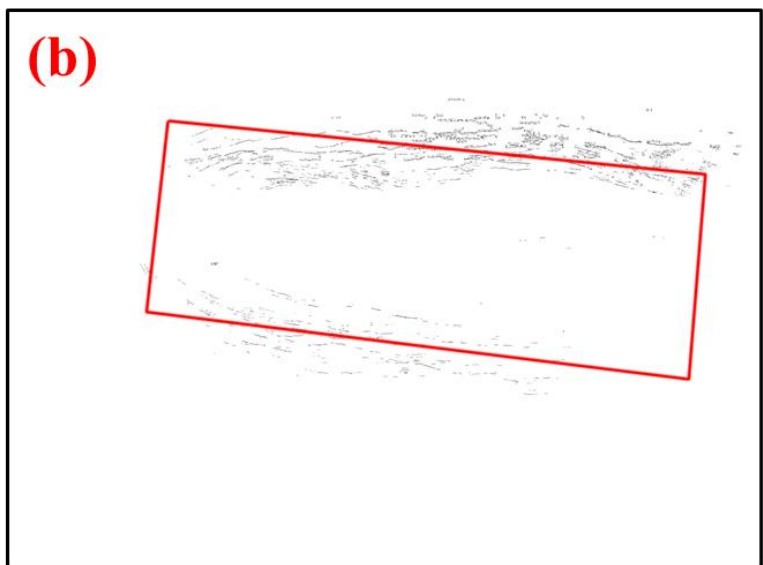

**Figure 6.** The crack extraction results of the UAV image: (**a**) original image, and (**b**) crack extraction results. The red frame is the coal mining face.

**Table 3.** The experimental results for moisture evaporation for loess soil under different crack densities in 24 h.

| Number | Crack Density | Crack Size | Initial Weight/g | Weight after 24 h-Evaporation | 24 h Moisture Evaporation |
|---|---|---|---|---|---|
| b1 | 0 | \ | 4483.1 | 4373.2 | 109.9 |
| b2 | 0.40% | 0.5 × 3.2 | 4497.5 | 4385.2 | 112.3 |
| b3 | 0.80% | 0.5 × 6.4 | 4468.4 | 4350.9 | 117.5 |
| b4 | 1.20% | 0.5 × 9.6 | 4479.2 | 4361.5 | 117.7 |
| b5 | 1.60% | 0.5 × 12.8 | 4490.5 | 4371.9 | 118.6 |
| b6 | 2.00% | 0.5 × 16 | 4492.9 | 4373.1 | 119.8 |
| b7 | 5.00% | 1.0 × 20 | 4492.9 | 4361.3 | 131.6 |
| b8 | 10.00% | 2.0 × 20 | 4481.7 | 4325.8 | 155.9 |

**Table 4.** The experimental results for moisture evaporation from sandy soil under different crack densities in 24 h.

| Number | Crack Density | Crack Size | Initial Weight/g | Weight after 24 h-Evaporation | 24 h Moisture Evaporation |
|---|---|---|---|---|---|
| a1 | 0 | \ | 4755.5 | 4608.5 | 147.0 |
| a2 | 0.40% | 0.5 × 3.2 | 4731.2 | 4582.6 | 148.6 |
| a3 | 0.80% | 0.5 × 6.4 | 4789 | 4633.2 | 155.8 |
| a4 | 1.20% | 0.5 × 9.6 | 4810.3 | 4651.4 | 158.9 |
| a5 | 1.60% | 0.5 × 12.8 | 4820.4 | 4660.9 | 159.5 |
| a6 | 2.00% | 0.5 × 16 | 4797.3 | 4635.5 | 161.8 |
| a7 | 5.00% | 1.0 × 20 | 4839.3 | 4671.5 | 167.8 |
| a8 | 10.00% | 2.0 × 20 | 4866.8 | 4686.5 | 180.3 |

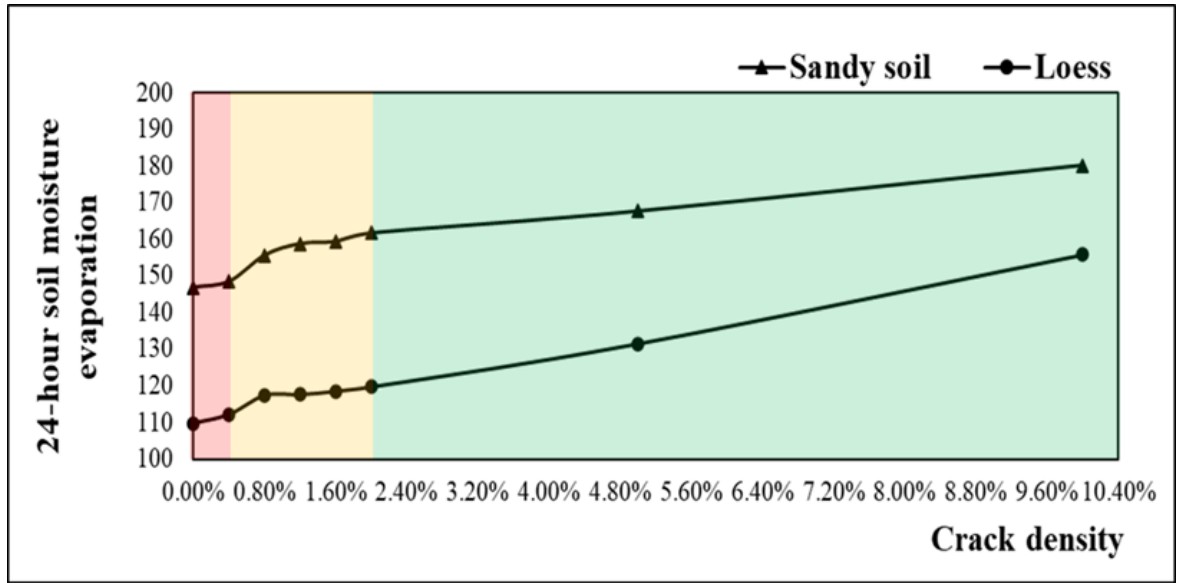

**Figure 7.** The results for soil moisture evaporation in 24 h from loess and sandy soils at the different crack densities.

**Table 5.** The results for the construction of grading criteria for the degree of crack development.

| Crack Development Degree | Slight Development | Moderate Development | Severe Development |
|---|---|---|---|
| Grading criteria (crack density) | <0.4% | 0.4–2.0% | >2.0% |

Based on the construction results of the grading criteria, the degree of crack development in the study area was evaluated based on grids, and the results are shown in Figure 8. The size of the study area image is 28,000 × 20,000 pixels. The size of the grid image is 400 × 400 pixels (approximately 5.2 m × 5.2 m). The total number of grids is 3500, and the number of grids with cracks is 472, accounting for 13.46% of the whole study area. From Figure 8, it can be seen that the study area mainly contains two major areas, the northern crack zone and the southern crack zone, in which the northern crack zone is significantly more developed than the southern crack zone. Table 6 shows the statistical results for the degree of crack development based on the grid, and the maximum crack grid density is 4.37%. The number of grids with slight development, moderate development, and severe development was 181, 257, and 35, respectively, accounting for 38.27%, 54.33%, and 7.40%. The study area is dominated by moderate development, the number of grids with severe

development is low, and they are mainly located in the northern crack zone of the study area.

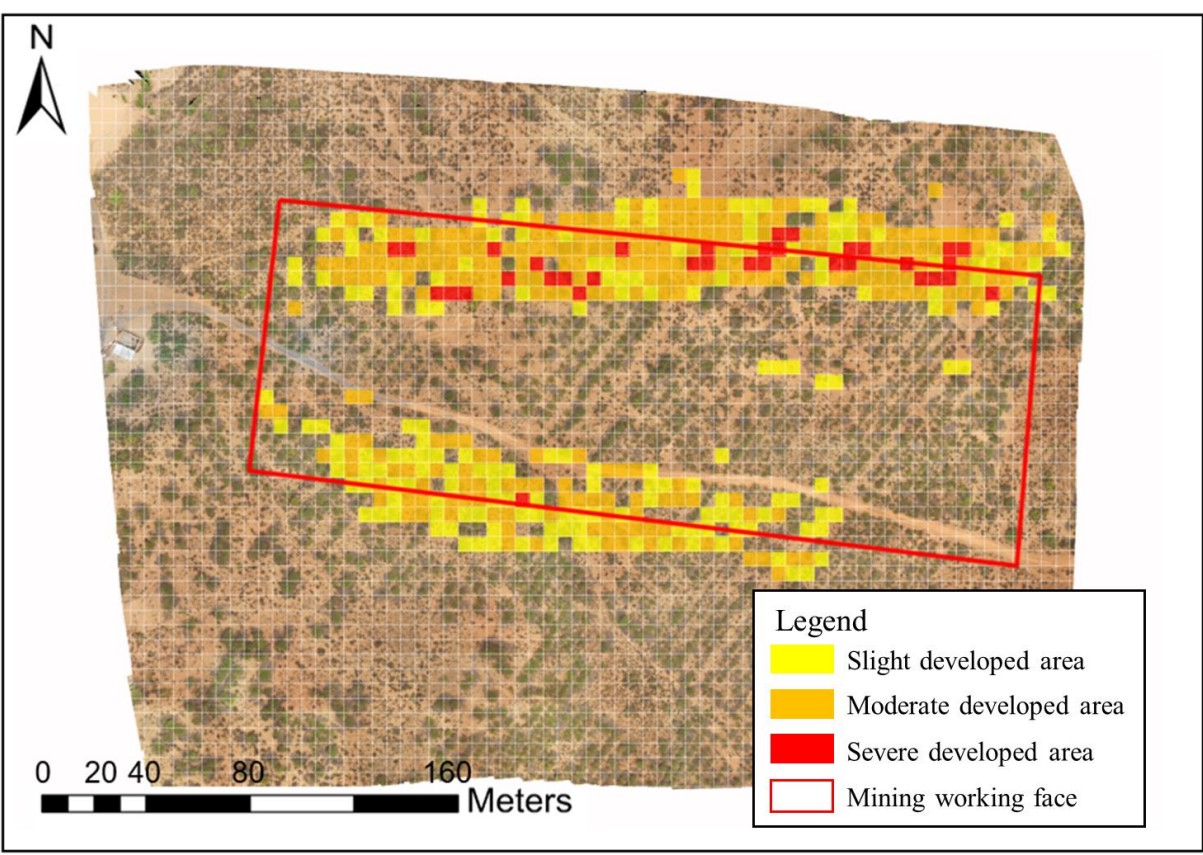

**Figure 8.** The results for soil moisture content measurements of loess and sandy soil at different distances from the crack.

**Table 6.** Statistical results for the degree of crack development based on grids.

| Crack Development Degree | Slight Development | Moderate Development | Severe Development |
|---|---|---|---|
| Crack density range | (2.01%, 4.37%) | (0.40%, 1.98%) | (0.01%, 0.39%) |
| Number of grids | 181 | 257 | 35 |
| Percentage | 38.27% | 54.33% | 7.40% |

*3.3. The Results for the Construction of Grading Criteria and Evaluation for Soil Damage Degree*

The closer the soil is to the crack, the greater the soil moisture evaporation and the more severely damaged the soil. Therefore, in this study, the soil moisture content at different distances from the crack was measured in the field to invert the degree of soil damage and then construct the grading criteria. Figure 9 shows the results of soil moisture content measurements for loess and sandy soil at different distances from the crack. It can be seen that the soil moisture content is low and does not vary significantly when the distance from the crack location is within 0.6 m. The soil moisture content gradually increases when the distance from the crack location is between 0.6 m and 1.2 m, indicating a gradual decrease in the effect of cracks on soil moisture evaporation. When it reaches 1.2 m, the change in soil moisture content is not obvious and the cracks have no significant effect on the evaporation of soil moisture. Therefore, the results for the construction of grading criteria for the degree of soil damage in this study are shown in Table 7. The soil within 0.6 m from the crack is considered an area with severe damage. The soil between 0.6

and 1.2 m from the crack is an area of slight damage. The soil more than 1.2 m away from the crack corresponds to an area of no obvious damage.

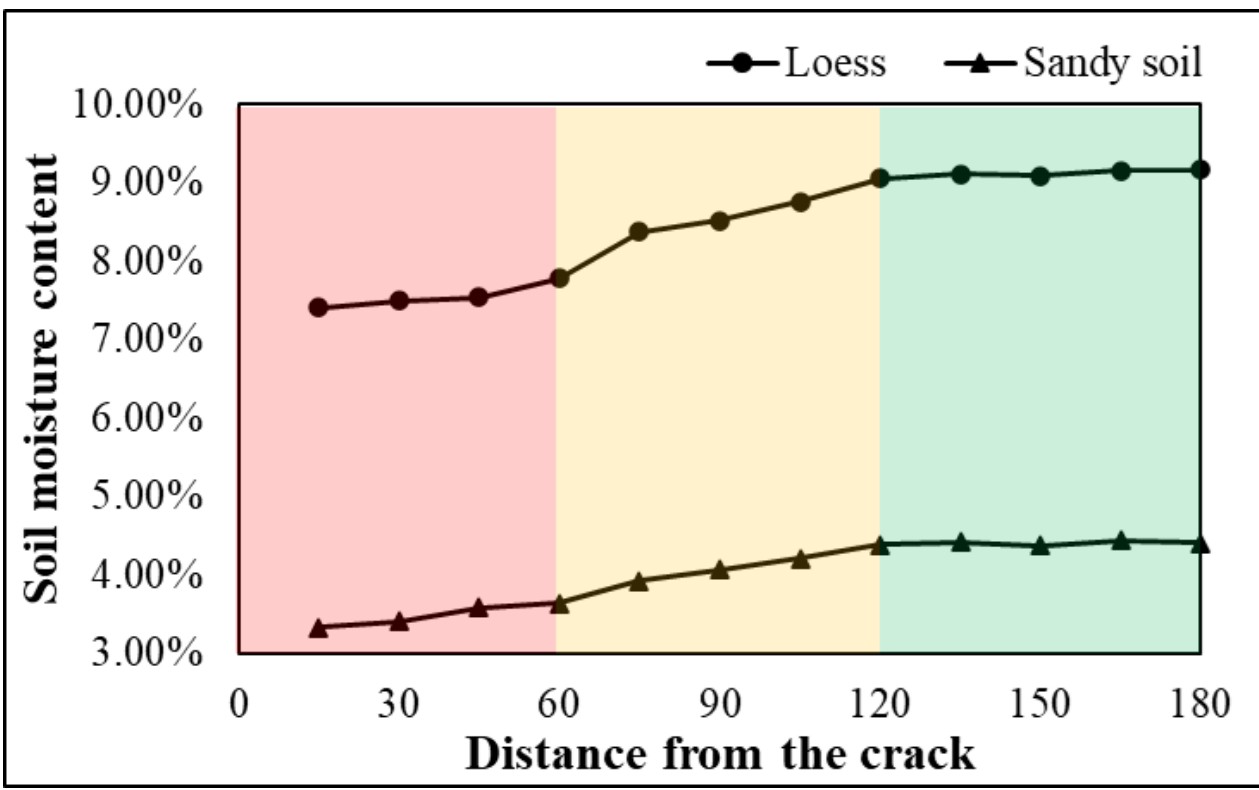

**Figure 9.** The results of soil moisture content measurements for loess and sandy soil at different distances from the crack.

**Table 7.** The results for the construction of grading criteria for the degree of soil damage.

| Soil Damage Degree | No Obvious Damage | Slight Damage | Severe Damage |
|---|---|---|---|
| Grading criteria (distance from the crack) | <1.2 m | 0.6–1.2 m | <0.6 m |

Based on the construction results of the grading criteria, the degree of soil damage in the study area was evaluated based on the buffer and the results are shown in Figure 10. Table 8 shows the statistical results of the degree of soil damage. It can be seen that the area of no obvious damage is 87,080.55 m$^2$, the area of slight damage is 3180.10 m$^2$, and the area of severe damage is 4436.12 m$^2$, accounting for 91.96%, 3.36%, and 4.68%, respectively, of the entire study area. The area of severe damage is higher than the area with slight damage. The reason for this phenomenon is that the distribution of cracks is more concentrated, especially in the northern crack zone. The cracks and damaged soil showed a striped pattern in the east-west direction.

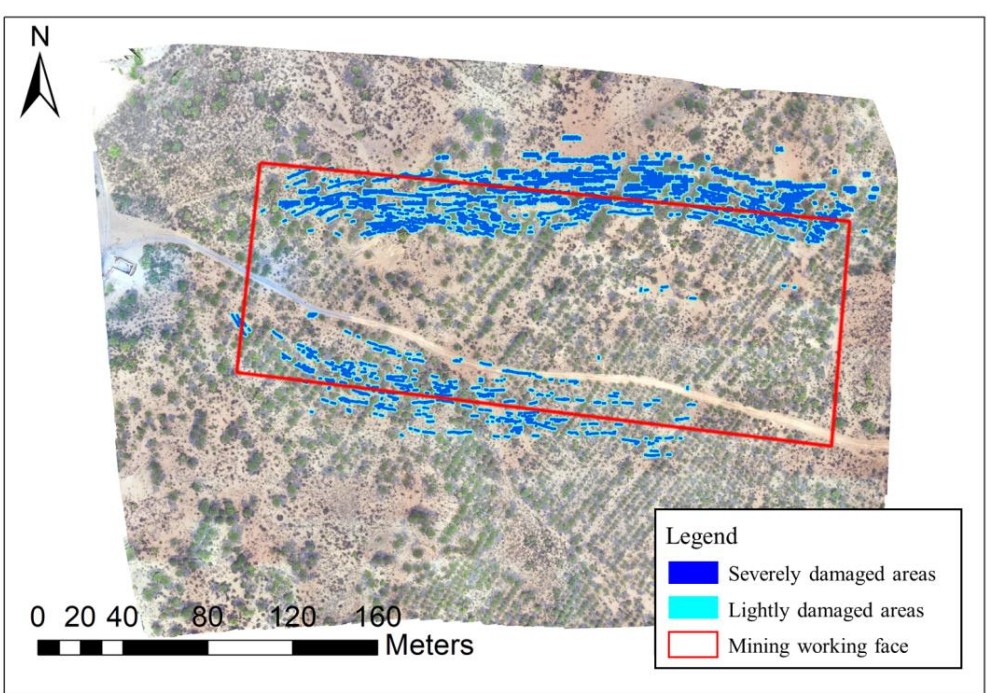

**Figure 10.** Evaluation results of the degree of soil damage based on the buffer.

**Table 8.** The statistical results of the degree of soil damage.

| Soil Damage Degree | No Obvious Damage | Slight Damage | Severe Damage |
|---|---|---|---|
| Area (m$^2$) | 87080.55 | 3180.10 | 4436.12 |
| Percentage | 91.96% | 3.36% | 4.68% |

## 4. Discussion

Surface cracks are an important cause of surface ecological damage in mining areas. Monitoring of surface cracks is the basic work of land reclamation and ecological restoration in mining areas [13]. Traditional monitoring methods cannot be used to efficiently obtain high-precision crack distribution information of large coal mining subsidence areas, which restricts the design of crack repair and land reclamation plans. With the rapid development of UAV remote sensing technology in recent years, the accuracy of crack extraction in large-scale coal mining subsidence areas has significantly improved. There is an increasing number of studies for crack extraction in mining areas based on UAV images [9,29,37]. This provides reliable data support for scholarly research regarding crack evaluation. This study evaluates the degree of both crack development and soil damage from two aspects based on UAV images. The aim is to design scientific crack repair and land reclamation plans by identifying the areas with more serious crack development and soil damage in the coal mining subsidence area and achieving accurate positioning of the crack location.

Regarding evaluation of the crack development degree, in this study, grading criteria are constructed and an evaluation method based on grids is proposed. Surface cracks caused by underground coal mining often appear to separate. This phenomenon is due to complex geological conditions, the presence of loose layers of overlying rock, and the vegetation cover on the surface. This problem cannot be solved better by morphological image processing methods such as hole filling and crack bridging, etc. Therefore, it is unreasonable to evaluate the degree of crack development through characteristic information such as the length and width of individual cracks. Moreover, the severity of crack development in different locations in the study area is not reflected. In this article, the crack development evaluation method based on the grid involves dividing the entire study area into many small grids with location information. Evaluation based on the crack density

of the grid can effectively reflect whether the crack development is serious at different locations in the study area. This will also facilitate crack repair based on accurate location information. According to the advice of crack management experts, a grid of approximately 5 m is more convenient for restoration measures. The final evaluation results show that the degree of crack development is significantly higher in the northern crack zone than the southern crack zone, which is consistent with the results of the field survey.

Regarding the evaluation of soil damage degree, in this study, grading criteria are constructed and an evaluation method based on the buffer is proposed. The surface of the coal mining subsidence area is mainly soil and vegetation. The direct ecological impact of cracks is soil damage, which, in turn, affects the growth of vegetation. Therefore, in addition to the evaluation of the crack development degree, evaluation of the soil damage degree is also necessary. The soil damage gradually decreases with increasing distance from the crack [34,35,37]. In this study, inversion of the changes in soil moisture content measured in the field were used to determine the degree of soil damage. We found that soil within 0.6 m was more affected by the cracks and that there was no significant damage to the soil after more than 1.2 m. This is in agreement with the findings of Zhang H.B. and Xu C.Y. [35,36]. Surface cracks affect the moisture content of soil [38,39] and the root system of vegetation [40]. This causes the occurrence of vegetation withering and increased soil erosion. Therefore, we should not only repair the cracks but also plant vegetation appropriately to reduce soil moisture evaporation and prevent soil erosion due to the effects of wind and sand. According to the results for the evaluation of soil damage, vegetation should be planted in areas with no obvious damage that are more than 1.2 m away from cracks. At the same time, soil within 1.2 m of cracks should be replenished with water.

Combining the evaluation results of the crack development degree and soil damage degree can form the basis for suggestions of scientific and effective measures for mine managers in the design of crack repair and land reclamation plans. For surface cracks in coal mining subsidence areas, a self-repair phenomenon exists [25] wherein internal dynamic cracks will appear to close with the advancement of underground coal mining workings, while slightly developed cracks can also be naturally repaired by wind and rainfall. Therefore, based on the evaluation results of the crack development degree, we can carry out repair measures such as filling only in the areas where cracks show moderate and severe development. Then, combined with the evaluation results of the soil damage degree, we can plant vegetation in areas more than 1.2 m away from the cracks while supplementing the soil within 1.2 m of cracks with water. Thus, the ecological restoration model of "natural restoration + crack filling + water supplementing + vegetation planting" is adopted to achieve crack restoration and land reclamation.

This study is the first attempt to establish a classification standard for the surface cracks caused by underground coal mining in the mining area and to establish a classification standard for the degree of soil damage caused by the impact of cracks. Compared with the existing research [24], which is evaluated by the nuclear density method, the grading basis of this study is based on the results of indoor simulation experiments and field measurements, which is more scientific and reasonable, and has more practical guiding significance for crack control and land reclamation. However, since the construction method of its grading standard is proposed for the first time and has not been supported by literature, further research is still needed in the future to obtain more scientific conclusions.

The method proposed in this study is mainly applicable to arid and semi-arid areas or areas with less vegetation. The accuracy of crack extraction will be reduced due to the interference of vegetation when crack extraction is carried out in areas with lush vegetation.

## 5. Conclusions

Based on the crack extraction results of UAV images, in this study, the degrees of both crack development and soil damage degree were evaluated. The grading index was constructed according to the effect of cracks on soil moisture through indoor experiments and

field measurements, and the entire study area was evaluated. The following conclusions were obtained:

(1) Crack density is used as an evaluation index and a grid of approximately 5 m is used as an evaluation unit for the construction of grading criteria for the crack development degree. The crack density is less than 0.4% for slight development, 0.4% to 2% for moderate development, and more than 2% for severe development.

(2) The cracks in the study area are mainly moderate, accounting for 54.33%, and only severe in 7.40% of areas, which are mainly located in the northern fracture zone of the study area. Significantly more severe cracks were found in the northern crack zone than the southern crack zone.

(3) The distance between the soil and the crack is the basis for the evaluation index for the soil damage degree. A distance within 0.6 m corresponds to an area with severe damage, between 0.6 and 1.2 m to slight damage, and more than 1.2 m to no obvious damage.

(4) The percentages of the severe damage area and slight damage area in the study area were 3.36% and 4.68%, respectively. The distribution of cracks was relatively concentrated, and the cracks and damaged soil showed a striped pattern in the east-west direction.

(5) By combining the evaluation results for the crack development degree and soil damage degree, the ecological restoration model of "natural restoration + crack filling + water supplementing + vegetation planting" was adopted to achieve crack restoration and land reclamation, with natural restoration for cracks in areas of slight development, crack filling for cracks in areas of moderate and severe development, water supplementation in soil areas within 1.2 m from the crack, and vegetation planting in soil areas more than 1.2 m from the crack.

The rapid development of UAV remote sensing technology has resulted in an ideal data source for the acquisition of crack feature information. Through scientific and reasonable evaluation of the degree of crack development and soil damage, it can provide supporting data and a theoretical basis for crack repair and land reclamation. However, the construction of the grading criteria is still somewhat subjective and needs to be studied in depth. Meanwhile, how to develop a more reasonable and effective land reclamation plan based on the evaluation results also requires further discussion and research.

**Author Contributions:** Conceptualization, F.Z.; methodology, Z.H.; software, Q.L.; validation, F.Z.; formal analysis, F.Z.; investigation, F.Z.; resources, Y.L. and F.Z.; data curation, F.Z.; writing—original draft preparation, F.Z.; writing—review and editing, Z.H.; visualization, F.Z.; supervision, F.Z.; project administration, Z.H.; funding acquisition, Z.H. All authors have read and agreed to the published version of the manuscript.

**Funding:** This research was funded by the Research and Demonstration of Key Technology for Water Re-sources Protection and Utilization and Ecological Reconstruction in Coal Mining Areas of Northern Shaanxi; the grant number is 2018SMHKJ-A-J-03.

**Data Availability Statement:** Not applicable.

**Acknowledgments:** Thanks go to the Institute of Land Reclamation and Ecological Reconstruction and Shaanxi Coal Group for their help in this research.

**Conflicts of Interest:** The authors declare no conflict of interest.

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
