# Peer review of "Evaluation of Surface Crack Development and Soil Damage Based on UAV Images of Coal Mining Areas"

_land, doi:10.3390/land12040774_

Round 1

Reviewer 1 Report

An interesting article showing the real deformations of the land surface caused by mining activities and resulting in the deterioration of the surface properties of the soil. Several elements in the text require correction.

Comments:

It should be made clear that the methods presented are for desert or sparsely vegetated areas. They cannot be used in areas with lush vegetation.

The problems of removing interactions with vegetation during crack extraction can be solved using the results of the research described in the articles:

DOI: https://doi.org/10.1109/TGRS.2021.3057272; https://doi.org/10.3390/rs12111733; https://doi.org/10.1016/j.isprsjprs.2021.02.006;

Deformations resulting from mining operations causing cracks in the surface of the terrain should be located in the zones of maximum tensile deformations, i.e. in specific areas outside the contours of mining fields. It may be worthwhile to show schematically the location of mining fields in the study area. It is also possible to check the correlation of the fracture zones obtained with the zones of maximum mining deformations. Thus, it is possible to predict the occurrence of crack zones on the surface of the ground before damage and degradation of the soil occurs.

Other:

- low quality of drawings no. 3 (picture at the bottom), 4 (markings a1, a2, etc.), 6b

- improve the contrast of Figures 3 and 6b

- figure numbering, figure no. 8 is missing - a mistake in numbering, figure no. 9 appears twice

It is not clearly defined how the crack density was determined, it can be deduced from the description and table 2 that it is the ratio of the crack area to the area of ​​the sampling area (20x20cm). I understand the density of cracks as the number of cracks on a given surface, so it may be worth emphasizing clearly how the authors understand this concept.

The sub-image size of 50×50 pixels was determined empirically, but what other dimensions were tested?

Author Response

Response to Reviewer 1:

Thank you so much for your comments regarding our manuscript entitled " Evaluation of surface crack development and soil damage based on UAV images of coal mining areas " by Fan Zhang et al. submitted to Land. We have revised the manuscript according to the reviewer’s comments. The detailed reply content is described in the word file

Reviewer 2 Report

1. The abstract needs to be rewritten to reflect the innovation of the research results.

2. The quality of pictures needs to be improved.

3. Language needs deep polish.

Author Response

Response to Reviewer 2:

Thank you so much for your comments regarding our manuscript entitled " Evaluation of surface crack development and soil damage based on UAV images of coal mining areas " by Fan Zhang et al. submitted to Land. We have revised the manuscript according to the reviewer’s comments.  The detailed reply content is described in the word file

Reviewer 3 Report

A brief summary: In the publication, the authors present an algorithm for the analysis of images obtained from a UAV for the study of crack development and soil damage in coal mining areas. Due to the rapid development of UAVs, this area of research is quite relevant and important. The proposed algorithm is quite new and interesting.

General concept comments: In general, the manuscript is written quite qualitatively and professionally, but despite this, it needs some major corrections. In the Introduction, problems with the appearance of cracks are analyzed in sufficient detail, but a critical analysis of their research methods is not carried out (see Specific comments 1). The Materials and Methods need clarification (see Specific comments 2-4). The obtained results are interesting but need clarification and confirmation (see Specific comments 5-6). The Discussion lacks a critical analysis of the obtained results (see Specific comments 7-8). The wording of the Results needs improvement. The References used by the authors are new and necessary. All the main technical parts of the publication are present.

Specific comments

1) Complement the introduction with a critical analysis of image analysis methods based on artificial intelligence and deep learning, emphasising crack analysis. Also, indicate the positive and negative sides of such methods.

2) The methodology involves several stages, but it is very difficult for an external reader to understand it, so I suggest making one general block diagram in which to present all steps from data acquisition to their analysis. It is also necessary to simplify the description of the methodology.

3) UAV images are the basis for conducting research. Therefore, it is necessary to describe in detail how they were obtained. Specify the number of reference and control points, and the method of determining their coordinates. Specify how the images were processed, and in which software.

4) The approximate location of the underground coal mines should be indicated on the map of the research region, this will help in the analysis of the results

5) The quality and accuracy of automatic surface crack extraction should be checked by other methods, including field observations. And a detailed critical comparative analysis must be presented in the publication.

6) The proposed grading criteria for the crack development degree and for the soil damage degree are insufficiently substantiated. It is necessary to confirm based on the analysis of literary sources.

7) The discussion describes the general problems associated with the appearance of cracks, instead, in the discussion, it is necessary to perform a critical analysis of the obtained results, establish the novelty and relevance of the methodology, determine the main advantages and disadvantages of the proposed solutions in comparison with analogues.

8) I also think that the discussion should answer the question of why you identified cracks in two lanes, and not in the entire territory. How is it related to the location of the underground coal mines?

9) The article is oversaturated with information about the damage of cracks and the processes of their formation (this is in the introduction, methodology, results and conclusions), which do not really relate to the purpose of the study. I suggest shortening this information a bit.

10) In the article, the authors mention the issue of reclamation and try to formulate certain proposals in this regard. I believe that this issue is quite complex and much more research is needed to formulate proposals. Therefore, I propose to remove questions related to reclamation from this article.

Author Response

Response to Reviewer 3:

Thank you so much for your comments regarding our manuscript entitled " Evaluation of surface crack development and soil damage based on UAV images of coal mining areas " by Fan Zhang et al. submitted to Land. We have revised the manuscript according to the reviewer’s comments.  The detailed reply content is described in the word file.

Round 2

Reviewer 3 Report

The authors took into account the given comments quite qualitatively and responsibly. This version of the manuscript looks logical, structured, and most importantly complete.